# Red Light Irradiation In Vivo Upregulates DJ-1 in the Retinal Ganglion Cell Layer and Protects against Axotomy-Related Dendritic Pruning

**DOI:** 10.3390/ijms22168380

**Published:** 2021-08-04

**Authors:** Kathy Beirne, Thomas J. Freeman, Malgorzata Rozanowska, Marcela Votruba

**Affiliations:** 1School of Optometry and Vision Sciences, Cardiff University, Cardiff CF24 4HQ, UK; kathybeirne1@gmail.com (K.B.); freemant5@cardiff.ac.uk (T.J.F.); rozanowskamb@cardiff.ac.uk (M.R.); 2Cardiff Institute for Tissue Engineering and Repair, Cardiff University, Cardiff CF10 3NB, UK; 3Cardiff Eye Unit, University Hospital of Wales, Cardiff CF14 4XW, UK

**Keywords:** retinal ganglion cells, photobiomodulation, neuronal degeneration, dendropathy, neuroprotection, retinal disease, OPA1, animal model, ADOA

## Abstract

Retinal ganglion cells (RGCs) undergo dendritic pruning in a variety of neurodegenerative diseases, including glaucoma and autosomal dominant optic atrophy (ADOA). Axotomising RGCs by severing the optic nerve generates an acute model of RGC dendropathy, which can be utilized to assess the therapeutic potential of treatments for RGC degeneration. Photobiomodulation (PBM) with red light provided neuroprotection to RGCs when administered ex vivo to wild-type retinal explants. In the current study, we used aged (13–15-month-old) wild-type and heterozygous B6;C3-Opa1^Q285STOP^ (Opa1^+/−^) mice, a model of ADOA exhibiting RGC dendropathy. These mice were pre-treated with 4 J/cm^2^ of 670 nm light for five consecutive days before the eyes were enucleated and the retinas flat-mounted into explant cultures for 0-, 8- or 16-h ex vivo. RGCs were imaged by confocal microscopy, and their dendritic architecture was quantified by Sholl analysis. In vivo 670 nm light pretreatment inhibited the RGC dendropathy observed in untreated wild-type retinas over 16 h ex vivo and inhibited dendropathy in ON-center RGCs in wild-type but not Opa1^+/−^ retinas. Immunohistochemistry revealed that aged Opa1^+/−^ RGCs exhibited increased nitrosative damage alongside significantly lower activation of NF-κB and upregulation of DJ-1. PBM restored NF-κB activation in Opa1^+/−^ RGCs and enhanced DJ-1 expression in both genotypes, indicating a potential molecular mechanism priming the retina to resist future oxidative insult. These data support the potential of PBM as a treatment for diseases involving RGC degeneration.

## 1. Introduction

Retinal ganglion cell (RGC) deterioration is a characteristic feature of neurodegeneration, not only in ocular conditions such as glaucoma and autosomal dominant optic atrophy (ADOA) [1,2] but also in diseases of the central nervous system, including Alzheimer’s, Parkinson’s, and Huntington’s diseases [3,4,5]. While the ubiquity with which retinal degeneration manifests in these conditions presents it as an attractive potential biomarker for degeneration in the brain, it also renders a treatment imperative. However, there is currently no available treatment for the vision loss associated with the deterioration of RGC dendrites [6].

ADOA is the leading cause of inherited optic neuropathy and presents as a progressive, bilateral loss in visual acuity beginning in early childhood [7]. The majority of ADOA patients express mutations in the OPA1 gene, resulting in an increase in reactive oxygen species (ROS) and mitochondrial dysfunction leading to cellular demise, particularly in cells such as RGCs, which both express the gene and have a high metabolic demand [8,9]. The heterozygous B6;C3-Opa1^Q285STOP^ (Opa1^+/−^) mouse model develops a loss of visual acuity beginning at 12 months of age [10], coinciding with progressive pruning of RGC dendrites [11]. Inner retinal pathology can eventually progress to severe degeneration of the ganglion cell layer [12,13]. Primary cultures of immunopanned RGCs from Opa1^+/−^ mice display decreased respiratory capacity due to impaired mitochondrial morphology and function [14]. This may explain the reported selective vulnerability of ON-center RGCs in these mice due to the high metabolic demand of their glutamatergic synapses compared to GABAergic synapses of OFF-centre RGCs [11].

Photobiomodulation (PBM) involves the use of red or near-infrared light as a medical intervention and has been proposed as a treatment for retinal degeneration due to its putative neuroprotective effects [15]. Irradiation with red light ameliorated retinal dysfunction and/or degeneration in a number of models of ocular dysfunction, including age-related macular degeneration [16,17], retinopathy of prematurity [18], diabetic retinopathy [19], age-related deterioration [20], retinal ischemia [21], and following physical or light-induced retinal damage [22,23]. PBM has also been tested as a therapeutic for retinal dysfunction in humans in both clinical trials for age-related macular degeneration (NCT02725762, NCT03878420) and diabetic retinopathy (NCT03866473) and smaller pilot studies with generally promising results [24,25,26,27,28]. However, there are currently no studies demonstrating the therapeutic potential of in vivo PBM to inhibit future RGC degeneration.

While the underpinning mechanism is not yet fully understood, it is widely considered to involve the enhancement of mitochondrial function [29]. The terminal enzyme in the electron transport chain, cytochrome C oxidase (CCO), is able to absorb light between 650–980 nm in wavelength [30,31]. This stimulates the photodissociation of nitric oxide, increasing its bioavailability as a signaling molecule while also enabling the capacity of CCO to bind molecular oxygen, thereby enhancing electron flux through the electron transport chain and ATP production [32]. It has been reported that photobiomodulation with near-infrared light induces a moderate increase in reactive oxygen species (ROS) production without increasing oxidative stress [33] and actually reduced ROS in oxidatively-stressed neurons [34]. This modest generation of ROS can activate redox-sensitive transcription factors such as DJ-1, Nrf-2, and NF-κB and initiate upregulation of antioxidant, detoxification, and pro-survival pathways [35,36].

The association between PBM and mitochondrial function implicates it as a putative treatment of RGC degeneration caused by mitochondrial pathology such as that found in ADOA patients. It has already demonstrated therapeutic benefits following rotenone-induced mitochondrial dysfunction [37,38]. Furthermore, we previously showed that ex vivo PBM treatment of retinal flat-mounts inhibited axotomy-induced dendropathy of retinal ganglion cells (RGCs) [39]. However, before PBM can be suggested as a potential treatment for RGC degeneration, it is first necessary to investigate whether that PBM can be effective when delivered in vivo and to senescent cells.

It is unlikely that any treatment could reverse the chronic, progressive RGC degeneration that Opa1^+/−^ mice exhibit throughout their lifetime. Therefore, it is imperative to develop a treatment able to prevent the onset of RGC degeneration. An acute model of RGC dendropathy, comparable to the chronic degeneration observed in aged Opa1^+/−^ mice, can be generated by severing the optic nerve, thereby axotomising the RGCs [40]. RGC dendropathy progresses over time following axotomy while the retinal explant is maintained in culture [39]. Evaluating the potential for an acute, in vivo treatment to inhibit axotomy-related dendropathy may indicate whether chronic administration of that treatment could deliver a preventative effect on the chronic RGC degeneration associated with ADOA, glaucoma, and Alzheimer’s disease. Meanwhile, it would also be interesting to discover whether any neuroprotective effect is evident in not only wild-type RGCs but also those with underlying pathology. There is evidence that PBM can trigger neuroprotection when administered prior to degenerative insult [41,42]. Thus, we hypothesized that in vivo PBM pretreatment could also inhibit dendropathy of RGCs following axotomy.

The purpose of the present study was to investigate both the in vivo therapeutic potential and putative mechanisms of PBM-induced neuroprotection of aged RGCs. We report that in vivo pretreatment with 670 nm light prevented the degeneration of WT RGCs over 16 h following axotomy. Loss of OPA1 resulted in increased nitrosative damage in the ganglion cell layer (GCL) alongside reduced NF-kB activation and upregulated DJ-1. However, despite the underlying nitrosative stress, light treatment was associated with the upregulation of DJ-1 in both genotypes, indicating a putative molecular mechanism.

## 2. Results

Aged (13–15-month-old) WT and Opa1^+/−^ mice were irradiated at the same time each day with 670 nm light for 88 seconds on five consecutive days. The mice were sacrificed 30 min following the final irradiation, and the eyes were processed for either immunohistochemistry (IHC) or RGC dendropathy analysis. In the latter group, retinal explants were dissected and stained with DiOlistic dyes either immediately (Figure 1A) or following 16 h in culture (Figure 1B).

### 2.1. Aged Opa1 RGCs Exhibit Significant Degeneration in Comparison with RGCs in Age-Matched Wild-Type Retinas

Sholl profiles, which display the number of dendritic intersections with concentric circles at increasing distances from the RGC soma (Figure 1C–E), revealed that aged Opa1^+/−^ RGCs were degenerated compared to age-matched WT controls. Sham-treated Opa1^+/−^ RGCs exhibited significantly reduced dendritic intersections at 20–80 µm from the soma (Figure 1C). As expected, the five-day 670 nm light treatment did not regenerate the lost dendrites, as the light-treated Opa1^+/−^ cells exhibited significant differences at 70, 90–150, and 180–200 µm from the cell body (Figure 1D). It is interesting that the sham- and light-treated cells exhibit degeneration at different points on the Sholl curves. However, there were no significant differences when comparing WT Sham to WT Light or Opa1 Sham to Opa1 Light (Figure 1E). Dendritic complexity was also quantified as three single measures which were consistent with the degeneration showed by the Sholl analysis of sham-treated cells. The area under the Sholl curve (AUC) (Figure 1F) was reduced by 18% (*p* < 0.05); the maximum peak of the Sholl curve was reduced by 17% (*p* < 0.01) (Figure 1G), and the total dendritic length was 436 μm shorter (18%) in the sham-treated Opa1^+/−^ group compared to the WT-Sham cells (Figure 1H). For the light-treated cells, significant degeneration was reported in the AUC (25% reduced) and total dendritic length (578 µm shorter, 24% reduced) (*p* < 0.05). However, the maximum Sholl peak comparison did not reach significance (*p* = 0.084).

### 2.2. 670 nm PBM Inhibited the Axotomy-Induced Dendropathy in Aged WT RGCs

Figure 2 depicts a significant loss of dendritic complexity in sham-treated WT mice after both 8- and 16-h ex vivo (Figure 2A). There were fewer dendritic intersections at 20, 50, and 60 µm from the soma after 8 h (*p* < 0.05) and at 20–100 µm (25%, *p* < 0.05) from the soma after 16 h ex vivo. The dendropathy exhibited by sham-treated RGCs all resulted in 26% significant reductions in the AUC, Sholl peak, and total dendritic length (617 µm) in the group that had been cultured for 16 h before staining (Figure 2G–I). Box and Whisker plots of the AUC, Sholl peak and total dendritic length are available in Appendix A. There was no significant dendropathy of light-treated WT RGCs measured either on the Sholl curves (Figure 2B) or the other quantifications of dendritic complexity (Figure 2G,H). While there were no significant differences when comparing the Sholl profiles of sham- and light-treated RGCs after 16 h in culture (Figure 2C), the AUC, Sholl peak, and dendritic length were all significantly greater in the light-treated group (Figure 2G–I), indicating that the 670 nm light had inhibited the axotomy-related dendropathy.

When considering the entire population of cells, sham-treated Opa1^+/−^ RGCs did not exhibit any changes over the duration of time spent in culture on the Sholl profiles (Figure 2D), AUC (Figure 2G), Sholl peak (Figure 2H), or dendritic length (Figure 2I). Light treated Opa1^+/−^ RGCs exhibited significant reductions in the number of intersections at 20 and 40 μm from the soma after 16 h in culture (Figure 2E, *p* < 0.05), but no significant differences in the other measures (Figure 2G–I). The two 16-h Opa1^+/−^ treatment groups did not show any differences in their Sholl profiles (Figure 2F).

Although the light treatment showed very promising effects in the WT RGCs, the lack of significant dendropathy over time post axotomy in the Opa1 sham group precluded the ability to measure an effect of PBM in the mutants.

### 2.3. PBM Did Not Inhibit Axotomy-Induced Dendropathy in the ON-Centre RGC Subpopulation

Rather than extend the duration of time spent in culture, which would increase the likelihood of apoptosis initiation and RGC cell death, we hypothesized that ON-center RGCs, which degenerate more rapidly than their OFF-centre counterparts in the Opa1^+/−^ mouse [11], would be more likely to show degeneration in the 16-h window and thus permit analysis of the light treatment in the mutant mice (Figure 3).

Sham-treated ON-center WT RGCs exhibit significant reductions from baseline in the number of dendritic intersections at 40, 50, and 60 µm (*p* < 0.05) from the soma after 8 h, and at 30–70 µm and at 80–90 µm (*p* < 0.05) from the soma following 16 h ex vivo (Figure 3A). In the light-treated WT RGC Sholl curve, there were fewer intersections at 90 µm from the soma after both 8- and 16-h post axotomy (*p* < 0.05, Figure 3B).

Further analysis of the ON-center RGCs from sham-treated WT mice mirrored the results from the entire population of cells. Following 16 h ex vivo, the AUC decreased by 28% (*p* < 0.01), the maximum Sholl peak decreased by 27% (*p* < 0.01), and the total dendritic length was reduced by 621 μm (27%, *p* < 0.01) compared to baseline (Figure 3G–I). Box and whisker plots of the data in Figure 3G–I are provided in Appendix A. This significant degeneration over time following axotomy was not observed in light-treated WT RGCs (AUC *p* = 0.116; maximum Sholl peak *p* = 0.099; dendritic length *p* = 0.147). When directly comparing the sham- and light-treated ON-center RGCs from WT mice after 16 h in culture, the Sholl profile of the light-treated cells revealed significantly more dendritic intersections at 20 µm and 30 µm (*p* < 0.05, Figure 3C). However, unlike in the total population, the AUC, Sholl peak, and total dendritic length were not greater in the light-treated cells (*p* = 0.445, 0.29, and 0.42, respectively).

Unlike the entire RGC population, the sub-population of ON-center RGCs from sham-treated Opa1^+/−^ mice exhibited significant changes in their Sholl profiles over the time spent in culture. Cells in the 16-h group had fewer dendritic intersections at 30 µm, 50 µm, and 60 µm from the soma (*p* < 0.05) (Figure 3D). This dendropathy during the 16-h following axotomy was also measured by significant reductions in the AUC, Sholl peak, and total dendritic length (all 31%, *p* < 0.05) (Figure 3G–I).

Light-treated ON-center Opa1 RGCs also exhibited changes in their Sholl profiles with fewer intersections at 20–40 µm from the cell body after 16 h ex vivo (Figure 3E). This manifested in the peak of the Sholl curve being significantly reduced by 16% (*p* < 0.05). The AUC (*p* = 0.251 and dendritic length *p* = 0.312 did not change over the 16 h in culture (Figure 3G,I). There were also no significant differences in the Sholl profiles, AUC (*p* = 0.229), Sholl peak (*p* = 0.184), or dendritic length (*p* = 0.204) when comparing sham- and light-treated RGCs at 16 h post axotomy (Figure 3F–I), indicating that even in the population of ON-center Opa1 RGCs which did undergo dendropathy following axotomy, there was no effect of the light treatment.

In summary, pretreatment with 670 nm light significantly inhibited degeneration in the total population of WT RGCs, as the AUC, Sholl peak, and dendritic length were all significantly greater in the light-treated cells compared to sham-treated after 16 h in culture following axotomy. Furthermore, while significant degeneration was observed over 16 h following axotomy in ON-center RGCs from both WT and Opa1^+/−^ sham-treated groups, it did not occur in WT RGCs that had been pre-treated with 670 nm light. This shows the ability of in vivo PBM pretreatment to provide protection against RGC dendropathy in aged WT, but not Opa1mice.

### 2.4. IHC Analysis of Molecular Mechanisms

#### 2.4.1. Aged Opa1^+/−^ Mice Display Enhanced Nitrosative Stress in the GCL

3-nitrotyrosine is a quantifiable marker of oxidative and nitrosative stress, as it is a biochemical footprint of the damage caused by reactive oxygen and nitrogen species such as peroxynitrite [43]. For this reason, we measured the level of 3-nitrotyrosine (3-NT) in the GCL of sham- and light-treated WT and Opa1^+/−^ mice (Figure 4A). We found no effect of 670 nm light on 3-NT levels in the GCL of aged WT or Opa1^+/−^ mice (main effect of treatment F = 0.539, *p* = 0.481), and no interaction between genotype and treatment (F = 0.021, *p* = 0.887), (Figure 4B). However, a significant main effect of genotype (F = 8.7, *p* = 0.016) showed that 3-NT was upregulated by 58% in the GCL of aged Opa1^+/−^ mice compared to WT controls. An independent t-test comparing the sham-treated WT and Opa1 groups also reported a significant increase in the mutant GCL (t = −2.97, *p* < 0.05), indicating that the reported effect was not influenced by the light treatment.

#### 2.4.2. Neither Light Treatment nor the Opa1^+/−^ Mutation Affected Nrf2 Expression or Activation in the GCL

The transcription factor Nrf2 is activated by oxidative stress [44], resulting in its translocation from the cytoplasm to the nucleus [45], while there is evidence that it can be activated by 670 nm light exposure [46]. Therefore, we examined its potential role in the molecular response to light treatment. Nrf2 staining was present throughout the retina, most noticeably in the GCL, IPL, and INL in each experimental group (Figure 5A). However, there was no statistically significant difference in the expression of Nrf2 overall in the GCL or in either the nuclei or cytoplasmic regions between sham-treated WT and Opa1^+/−^ mice in response to 670 nm light exposure (Figure 5B–D). The nuclear to cytoplasmic ratio was calculated as a proxy measure of Nrf2 activation and also did not show any differences between the groups (Figure 5E). There were also no statistically significant differences in any of these measurements between sham-treated WT and Opa1^+/−^ mice or in either genotype in response to 670 nm light exposure.

#### 2.4.3. Aged Opa1^+/−^ Retinae Exhibited Reduced NF-kB Activation in the GCL, which was Upregulated by 670 nm Light Treatment

NF-kB, similar to Nrf2, is activated by reactive oxygen and nitrogen species and has also been implicated in the molecular mechanism of PBM [35]. The level of activation may be quantified by the nuclear: cytoplasmic ratio of NF-kB, as translocation to the nucleus occurs in response to ROS and initiates the transcription of genes involved in the regulation of the inflammatory response and cytoprotection [47].

NF-kB staining was present in the GCL of all experimental groups (Figure 6A). There were no main effects of either 670 nm light treatment or the Opa1 mutation on the overall expression of NF-κB in the GCL (Figure 6B). However, there was a significant interaction between treatment and genotype (F = 4.76, *p* = 0.05). Simple main effects analysis revealed that light treatment was significantly associated with enhanced NF-κB expression in the GCL of Opa1 mutant mice (F = 5.37, *p* = 0.039). There were no significant effects detected for fluorescence measured in either the nuclei or the cytoplasm. The nuclear: cytoplasmic ratio, representing the extent of NF-κB activation, showed a significant main effect of light treatment (Figure 6C–E), with the treatment associated with greater nuclear translocation (F = 6.25, *p* = 0.028). There was also a significant interaction between genotype and treatment (F = 5.15, *p* = 0.042) and simple main effects analysis revealed 22% lower nuclear translocation in sham-treated Opa1 mutants compared to sham-treated WT mice (F = 7.16, *p* = 0.02). However, nuclear translocation of NF-κB was increased in Opa1 mice treated with 670 nm light compared to sham-treated mutants (F = 10.56, *p* = 0.007).

#### 2.4.4. 670 nm Light Treatment Upregulated DJ-1 Expression in the IPL

Following the observed changes in 3-NT and NF-κB, we also investigated the transcription factor DJ-1, which is upregulated by increased oxidative stress and can activate NF-κB in its cellular cascade to promote cell survival [48]. The expression of DJ-1 was investigated by IHC (Figure 7A) and was found to be upregulated by 25% in the GCL of Opa1^+/−^ mice compared to WT (Figure 7B, *p* < 0.05). There was also a 35% increase in DJ-1 expression in the GCL in response to 670 nm light treatment in retinas from WT mice (*p* < 0.05), but not Opa1^+/−^ mice (*p* < 0.549). However, examination of the IPL, which contains the RGC dendrites, revealed a significant main effect of 670 nm light treatment, with a 23% upregulation in DJ-1 expression detected in light treated retinas (F = 5.0, *p* < 0.05, Figure 7C).

In summary, IHC analysis revealed that Opa1^+/−^ mutation was also associated with increased levels of 3-NT in the GCL. Furthermore, NF-kB activation was significantly reduced in the GCL of Opa1^+/−^ mice compared to WTs, while DJ-1 expression was upregulated in these mutants. The 670 nm light treatment resulted in increased expression and nuclear translocation of NF-kB in the GCL Opa1^+/−^ mice. Finally, both genotypes displayed increased DJ-1 expression in the IPL in response to the light treatment.

## 3. Discussion

Red and near-infrared light has been proposed as a treatment for a variety of retinal diseases. In our study, we assessed its potential to inhibit RGC dendropathy when administered prior to the degenerative insult of severing the optic nerve, which axotomizes the cells. Our results showed that aged (13–15-month-old) WT RGCs exhibited dendropathy over 16 h ex vivo post axotomy. This was inhibited by five days of 670 nm light treatment administered in vivo, prior to sacrifice and the severing of the optic nerve—as shown by the light-treated RGCs having significantly greater dendritic complexity after 16 h in culture than sham-treated cells. These results were comparable to our previous study in which light treatment was administered ex vivo, post axotomy directly to retinal explants taken from 9-week-old mice [39]. Taken together, these results show that irradiation with this wavelength is able to prime RGCs to become resilient to future insult, as well as inhibit degeneration following damage to the optic nerve.

As light treatment ameliorated dendropathy in healthy young and aged WT RGCs, we assessed its therapeutic potential in a mouse model of ADOA. For PBM to be a putative therapeutic to inhibit RGC degeneration, it is important to elucidate whether the neuroprotective effect is restricted to healthy cells or is able to be transduced in diseased RGCs with underlying mitochondrial pathology. Aged Opa1^+/−^ mice exhibited reduced dendritic complexity across the whole population of RGCs compared to WT mice, in contrast to Williams et al. (2010), who reported selective degeneration of ON-center RGCs [11].

The Opa1^+/−^ RGCs did not exhibit dendropathy over the course of 16 h post axotomy, potentially due to the fact that some of the cells most prone to degeneration had already undergone dendropathy prior to sacrifice. Extending the time-course beyond 16 h would introduce potential confounds caused by the initiation of apoptosis in the environment and cell death, reducing the yield of cells labeled by the DiOlistic staining. Therefore, we selected the ON-center sub-population of RGCs, which is postulated to be more susceptible to dendropathy due to its higher metabolic demand and therefore was more likely to exhibit axotomy-related dendropathy within the same time frame shown by the WT cells [11]. While RGC dendropathy was exhibited by the sham-treated ON-center Opa1^+/−^ 16 h ex vivo, it was also evident in the light-treated group, indicating that the axotomy-related dendropathy was not inhibited by the 670 nm pretreatment.

Considering the neuroprotective effect of the PBM pretreatment was restricted to WT cells, immunohistochemical analysis of the GCL was performed to elucidate underlying molecular effects exhibited in the mutants and how this may influence a putative PBM mechanism. The present study reports, for the first time, nitrosative damage alongside RGC degeneration in retinas of aged Opa1^+/−^ mice. Increased 3-NT staining in mutant retinas indicated excessive damage caused by peroxynitrite, the highly reactive product of the superoxide and nitric oxide interaction that is able to oxidize tyrosine amino acid residues [43]. OPA1 is heavily involved in mitochondrial maintenance, and its loss leads to inefficient oxidative phosphorylation and elevated ROS generation [14,49]. There was no effect of light treatment on 3-NT levels in the GCL of mutant mice. Red light PBM has previously been associated with the alleviation of nitrosative stress in the CNS [50,51]. However, these studies did not assess light treatment in an in vivo model of chronic mitochondrial dysfunction, and so it is unsurprising that the short course of 670 nm PBM treatment did not reverse the nitrosative damage that had already occurred.

Analysis of transcription factors sensitive to oxidative stress and previously associated with PBM provided the opportunity to assess mechanistic aspects of light treatment. DJ-1 upregulation was observed in the GCL of WT retinas and the IPL in both WT and Opa1^+/−^ mice following light treatment. While the transient release of ROS induced by PBM has been associated with activation of antioxidant transcription factors [52], to our knowledge, DJ-1 has not previously been directly implicated. Recent studies have concluded that DJ-1 is vital for mitochondrial function at the synapse, and its expression enables neurite outgrowth via the ATP synthase protein [53,54], supporting our data linking DJ-1 upregulation with neuroprotection of RGC dendrites. Other recent studies have demonstrated the neuroprotective role of DJ-1 in the retina, as loss of the gene is associated with retinal dysfunction and degeneration through reduction of retinal capacity to resist oxidative insult [55,56]. Meanwhile, upregulated expression of DJ-1 improved mitochondrial function, reduced ROS production, and inhibited apoptosis of retinal pericytes in an experimental model of diabetic retinopathy [57]. Therefore, the present study implicates DJ-1 upregulation as a potential mechanism underpinning the increase in mitochondrial efficiency, decrease in ROS generation, and neuroprotection associated with PBM.

Although Opa1^+/−^ retinas displayed upregulated DJ-1 in the IPL associated with light treatment, this was not observed in the GCL, which contains the cell bodies of RGCs. This was probably due to the fact that DJ-1 was already upregulated due to its role as a sensor of redox homeostasis in the mammalian retina, whereby it is upregulated in oxidative conditions, thereby limiting its capacity to respond to further oxidative signals [58]. Further evidence of the underlying oxidative environment found in Opa1^+/−^ retinas which may have contributed to the lack of neuroprotective response to PBM is displayed by the fact that NF-κB activation was decreased. NF-κB has a complex relationship with oxidative stress, as ROS can both induce and inhibit its activation [59]. Wu et al. demonstrated that while a transient increase in ROS stimulated the NF-κB pathway, sustained oxidative stress resulted in attenuated NF-κB activation due to impaired proteasome function, restricting the cell’s capacity to recover [60]. This may explain the reduced NF-κB activation aged mutant retinas observed in the present study, as they develop a chronic oxidative environment as shown by the increase in 3-NT.

The observed increase in NF-κB activation associated with light treatment is supported by previous studies, which indicate that PBM induces NF-κB activation through a transient increase in ROS [33,35]. This shows that that Opa1^+/−^ cells in the GCL were able to respond to 670 nm light but that this did not translate to inhibition of dendropathy following axotomy. The retinas in our study were fixed 30 min following the final PBM treatment. Therefore, it is probable that the observed amelioration of NF-κB activation did not persist over the 16 h ex vivo. We did not observe any changes in Nrf2, which was surprising because, like NF-κB, its activation has previously been associated with both oxidative stress and PBM [44,46].

While 670 nm light pretreatment over a period of five days was not expected to reverse the RGC degeneration or nitrosative damage in aged Opa1^+/−^ retinas, it also did not inhibit RGC degeneration following axotomy in these mutants. Together, these results support the use of red light as a preventative, rather than regenerative, treatment administered to healthy RGCs in order to manifest a neuroprotective effect against further dendropathy. This would be most efficacious for young people with early diagnoses of RGC diseases such as glaucoma and ADOA to prevent further visual decline.

A longer-term light treatment commencing at a younger age, prior to RGC deterioration would examine whether the inhibition of RGC dendropathy following axotomy reported in the present study translates to preventing the chronic degeneration observed in aged Opa1^+/−^ retinas. Further, as the pruning of the ON-center RGCs coincides with a deterioration in visual acuity in the Opa1^+/−^ mouse, it would be interesting to test whether light treatment could also prevent the associated visual decline. If red light therapy is able to inhibit the age-dependent RGC degeneration and visual impairment in Opa1^+/−^ mice, it may have potential as a future treatment for ADOA and other optic neuropathies involving RGC pathology.

In summary, 670 nm light pretreatment inhibited the axotomy-related RGC dendropathy when administered in vivo to aged WT mice prior to severing the optic nerve. However, neuroprotection was not observed in aged Opa1^+/−^ mice, indicating that the putative therapeutic effect did not translate to cells with underlying pathology and would need to be administered prior to the onset of RGC degeneration. Confirmation of the neuroprotective potential of red light administered in vivo prior to degenerative insult was important evidence that PBM may be effective as a preventative treatment to inhibit RGC degeneration.

## 4. Materials and Methods

### 4.1. Animals

Breeding, maintenance, and all experimental procedures in this study were carried out in accordance with the UK Animals (Scientific Procedures) Act 1986. Opa1^+/−^ and WT mice were kept in a 12-h light (10 lux) -dark cycle with food and water available *ad libitum* and aged until 13–15 months old. Generation of the mutant strain, B6; C3-Opa1^Q285STOP^ (Opa1^+/−^) has been described in detail elsewhere [10]. The model expresses a protein-truncating mutation in exon 8 of Opa1 that leads to Opa1 haploinsufficiency.

### 4.2. 670 nm Light Treatment

Animals were divided into four experimental groups: wild-type sham-treated (15 mice), WT light-treated (16 mice), Opa1^+/−^ sham (16 mice), and Opa1^+/−^ light (15 mice). Light-treated mice were exposed to 4.4 J/cm^2^ of 670 nm light in each eye delivered over 88 s at an irradiance of approximately 50 mW/cm^2^ from two WARP 10 light sources (Quantum devices, Newark, OH, USA), after which the devices automatically turned off with a delay timer preventing their reactivation (Figure 8A). The device, irradiance, and total dose matched those used to positive effect in clinical studies of patients with AMD and diabetic macular edema with promising effects in the TORPA (NCT00940407) and LIGHTSITE 1 (NCT02725762) trials [24,25]. Irradiance level at the cornea was measured by a JETI Specboss 1201 spectroradiometer (JETI Technische Instrumente GmbH, Jena, Germany). All mice were adapted to ambient light for 30 min prior to treatment. Sham-treated animals were subjected to the same handling with the light sources turned off (Figure 8B). Mice were treated for five consecutive days between 9 a.m. and 10 a.m. each day.

On the fifth day, the final light or sham treatment was administered 30 min prior to sacrifice and retinal explant dissection. 30 min was chosen as the delay between the final treatment and axotomy as it has been shown that nitric oxide-induced gene expression peaks 30 min following exposure to nitric oxide in cultured cells [61].

### 4.3. Preparation of Eyes for Analysis

Following sacrifice, eyes were taken for either dendritic processing using DiOlistic labeling or IHC. For IHC analysis, eyes were placed in 4% paraformaldehyde and left overnight at 4 °C. It was then transferred to a 5% sucrose solution for 1 h and subsequently, 30% sucrose overnight. The eyes were then embedded in OCT, frozen in melting isopentane, and stored at −80 °C, ready for IHC (see below). For DiOlistic staining and dendropathy analysis, eyes were enucleated, axotomising the RGCs, prior to retinal dissection and preparation of explants as described previously [11]. Retinas from each group were assigned to three sub-groups and underwent DiOlistic labeling either immediately or following delays of 8 or 16 h after axotomy.

### 4.4. DiOlistic Labeling and Analysis of RGCs

Tungsten particle bullets (1.7 µm; Bio-Rad, Hercules, CA, USA) coated with 1,1′-dioleyl-3,3,3′,3′-tetramethylindocarbocyanine methanesulphonate (DiO) and 3,3′dihexadecyloxacarbocyanine perchlorate (DiI) lipophilic dyes (Invitrogen, Waltham, MA, USA) were prepared as previously described [11]. The protocol for the DiOlistic labeling, imaging, and analysis of RGCs has also been described [39]. Briefly, retinal explants were shot once with dye-covered tungsten bullets at 100 psi using a Helios gene gun (Bio-Rad) through a 3 μm pore size, high pore density cell culture insert (Millipore, Billerica, MA, USA) 3 cm from the retina. Following labeling, retinal explants were transferred to pre-warmed Neurobasal A and incubated (5% CO_2_, 37 °C) for 30 min, allowing diffusion of the dye within the cells before fixation with 4% paraformaldehyde. Finally, explants were stained with TO-PRO nuclear stain, mounted with Prolong Gold Antifade Reagent (Invitrogen, Waltham, MA, USA) and imaged with a Zeiss LSM 510 confocal microscope (Carl Zeiss Vision UK Ltd., Birmingham, UK) using a 20× (0.8 NA) objective lens.

The random labeling technique and large sample sizes have previously been shown to negate any potential impact of eccentricity on morphological comparisons [62]. RGCs were identified as having their soma located in the ganglion cell layer, their dendrites stratifying across the IPL with a distinct axon projecting towards the optic nerve. ON-center RGCs can be recognized due to their ramification in sub-lamina b of the inner plexiform layer (IPL), compared to OFF-centre RGCs in which the dendrites project towards the outer part of the IPL—sublamina a (see Williams et al. 2010 for more detailed characterization). The ‘Simple Neurite Tracer’ Fiji plugin [63] was used to reconstruct a 3D image of each cell by an experimenter blind to the experimental conditions before analysis by the automated Sholl analysis Fiji plugin. Sholl analysis [64] gives a quantitative measure of dendritic complexity, counting the number of dendritic intersections with concentric rings placed at 10 µm intervals from the soma to the most peripheral dendrite.

### 4.5. Immunohistochemistry

Sagittal sections of 10 µm thickness were cut from fixed-frozen eyes using a Leica 3050 S cryostat and allowed to air dry overnight before storage at −20 °C. Sections were adjusted to room temperature before washing in 0.1 M PBS and blocking with 10% fetal calf serum (Sigma–Aldrich, Gillingham, UK) in PBS for 1 h at room temperature. Primary antibodies were diluted in 5% FCS in PBS at the dilutions stated and incubated with the sections overnight at 4 °C. Rabbit anti-PARK7/DJ-1 (1:500, cat no. ab18257; Abcam Ltd., Cambridge, UK), rabbit anti-Nrf2 (1:100, cat no. ab31163; Abcam Ltd., Cambridge, UK), rabbit anti-nuclear factor kappa-light-chain-enhancer (NF-kB) p65 (phosphor S536) (1:150, cat no. ab86299; Abcam Ltd., Cambridge, UK), rabbit anti-nitrotyrosine (1:200, cat no. N0409; Sigma-Aldrich, UK). The secondary antibody goat anti-rabbit conjugated to Alexa Fluor 488 (1:500, cat no. ab150077; Abcam Ltd., Cambridge, UK) was diluted in 5% FCS in PBS, applied to sections and left for 2 h at room temperature. Sections were then stained with Hoechst 33342, trihydrochloride, trihydrate diluted to 1:1000 in PBS for 5 min at room temperature before cover slipping with Prolong Gold Antifade Reagent (Invitrogen, Waltham, MA, USA). Sections selected for no primary and no secondary antibody controls were subjected to identical treatment to the test samples, with the exception of the addition of 5% FCS in 0.1 M PBS in place of the primary and secondary antibody, respectively.

### 4.6. Imaging and Fluorescence Quantification

Images were acquired using a Zeiss LSM 510 confocal microscope (Carl Zeiss Vision UK Ltd., Birmingham, UK) using 20× (0.8 NA) and 60× (1.4 NA) objective lenses, 488 nm excitation, and BP 500–530 nm emission filter for Alexa Fluor 488 and 350 nm excitation and BP 411–480 nm for Hoechst 33342. Three sections were used from each mouse, with one image taken per section. All images were acquired with the same gain, intensity, and exposure time.

Fluorescence quantification was performed using Fiji software [65] by an experimenter blind to the conditions of each group. RGB images were converted into binary mode, and the green and blue channels were split. The green channel was selected, a rectangular box of fixed area was placed on the layer of interest, and the mean grey value was calculated, allowing quantification of the average fluorescence in the GCL and IPL. To measure the nuclear: cytoplasmic ratio of fluorescent markers in the RGCs, first, the integrated density of RGC nuclei was calculated. The nuclei were traced in the blue channel, and the integrated density value was measured in the corresponding RGC nuclei regions in the green channel. This was subtracted from the integrated density for the GCL to calculate the integrated density values for the RGC cytoplasm. The ratio between the integrated density of the nuclear and the cytoplasmic regions of the GCL was then calculated.

### 4.7. Statistical Analysis

Data processing was carried out using Microsoft Excel (Office 2016), and statistical analysis was performed using IBM SPSS Statistics for Windows, Version 25.0 (IBM, Armonk, NY, USA) or JASP (version 0.14.1.0) software. The Shapiro–Wilke test was performed to check for normality in the data. Data sets with non-normal distributions were analyzed by the non-parametric Mann Whitney *U*-test, while normally distributed data were analyzed using independent *t*-tests or analysis of variance (ANOVA). *p*-values of less than 0.05 were considered statistically significant.

## Figures and Tables

**Figure 1 ijms-22-08380-f001:**
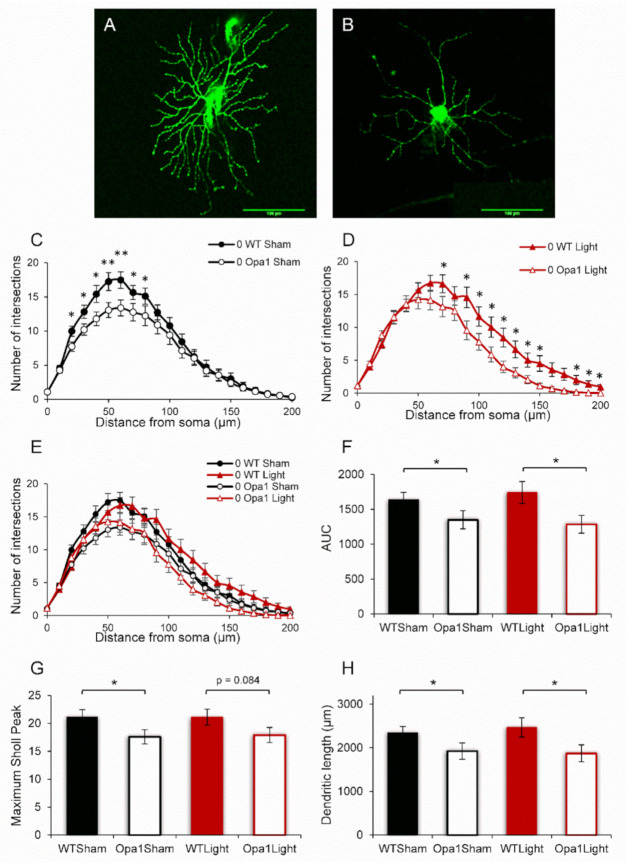
Aged Opa1^+/−^ RGCs display degenerated dendritic complexity. Representative images of (**A**) a DiOlistically stained healthy WT RGC and (**B**) one that has undergone dendropathy. Scale bar measures 100 μm. (**C**,**D**) Sholl profiles of both sham- and light-treated aged Opa1^+/−^ RGCs revealed significant degeneration compared to age-matched WT cells. (**E**) No effects of the light treatment were observed on either the WT or Opa1^+/−^ baseline dendritic complexity. (**F**) The area under the curve (AUC) expresses dendritic complexity as a single number and was significantly decreased in Opa1 RGCs. Dendritic complexity was also quantified by the maximum peak of the Sholl curve (**G**) and the total dendritic length (**H**). Mann-Whitney U-test * *p* < 0.05, ** *p* < 0.01. WT Sham *n* = 19 cells from 10 mice; Opa1 Sham *n* = 27 cells, 11 mice; WT Light *n* = 21 cells, 10 mice; Opa1 Light *n* = 20 cells, 8 mice.

**Figure 2 ijms-22-08380-f002:**
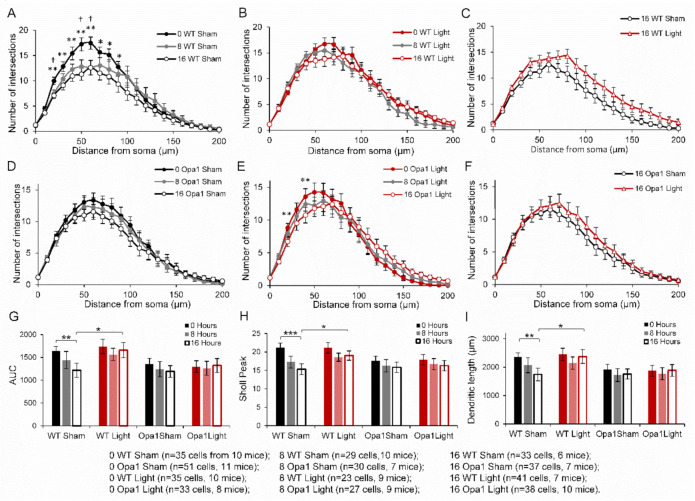
In vivo 670 nm pretreatment inhibited the axotomy-related dendropathy that occurred in sham-treated WT over 16 h following axotomy. Sholl profiles of (**A**) sham-treated aged WT RGCs showed significant degeneration over time post axotomy. (**B**) no effects were observed in light-treated WT RGCs. (**C**) Direct comparison of sham vs. light treatment following 16 h ex vivo. WT RGCs; (**D**) Sham- and (**E**) light-treated Opa1^+/−^ RGC Sholl curves showed no significant degeneration ex vivo. (**F**) a direct comparison of the two 16-h Opa1^+/−^ groups showed no significant effects. RGC dendritic complexity was also quantified by (**G**) the area under the Sholl curve (AUC); (**H**) average maximum peak of Sholl curve; and (**I**) average total dendritic length. All measures revealed significant degeneration over 16 h in the sham-treated WT RGCs, which was inhibited by the light treatment, as shown by the 16-h WT light group having significantly greater dendritic complexity. Daggers and stars indicate a statistically significant reduction in the number of dendritic intersections from 0 h to 8- and 16-h, respectively. * *p* < 0.05, ** *p* < 0.01, ****p* < 0.001, † *p* < 0.05; Mann-Whitney *U*-test. Error bars represent SEM.

**Figure 3 ijms-22-08380-f003:**
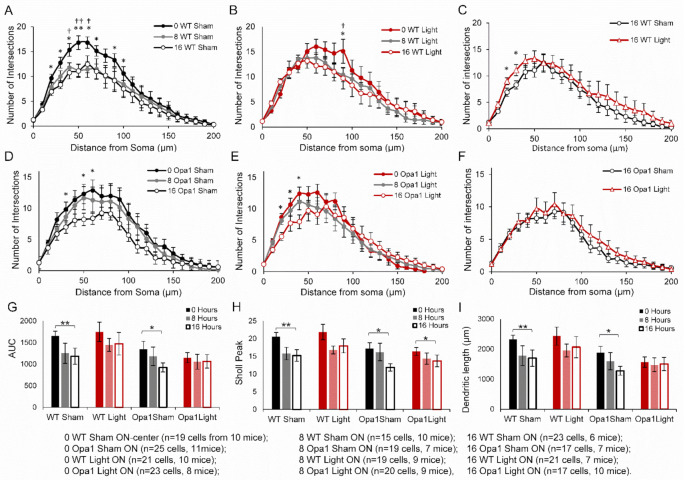
Sham-treated aged ON-center WT and Opa1^+/−^ RGCs exhibited significant axotomy-dendropathy while light-treated RGCs did not. (**A**) There was significant degeneration at multiple points along the Sholl curve for sham-treated WT RGCs at 8- and 16-h ex vivo post axotomy in sham-treated WT RGCs. (**B**) Sholl profiles from light-treated ON-center RGCs revealed a significant degeneration at 90 μm at 16 h post axotomy. (**C**) Direct comparison of the 16-h sham- and light-treated RGC Sholl curves showed a greater number of dendritic intersections at 20 and 30 μm in the light-treated group. (**D**) Sham- and (**E**) light-treated Opa1^+/−^ RGC Sholl profiles showed degeneration after 16 h ex vivo. (**F**) Sholl profiles comparing the sham- and light-treated ON-center Opa1 RGCs after 16 h ex vivo revealed no effect of light treatment. Quantification of RGC dendritic complexity by (**G**) the area under the Sholl curve (AUC), (**H**) average maximum peak of Sholl curve, and (**I**) average total dendritic length all revealed significant degeneration in the retinal explants of both sham-treated groups over 16 h. Light-treated WT cells did not exhibit significant degeneration, but the maximum Sholl peak was reduced in the light-treated Opa1 group. Daggers and stars indicate a statistically significant reduction in the number of dendritic intersections from 0 h to 8- and 16-h, respectively. * *p* < 0.05, ** *p* < 0.01, † *p* < 0.05, †† *p* < 0.01; Mann-Whitney *U* test. Error bars represent SEM.

**Figure 4 ijms-22-08380-f004:**
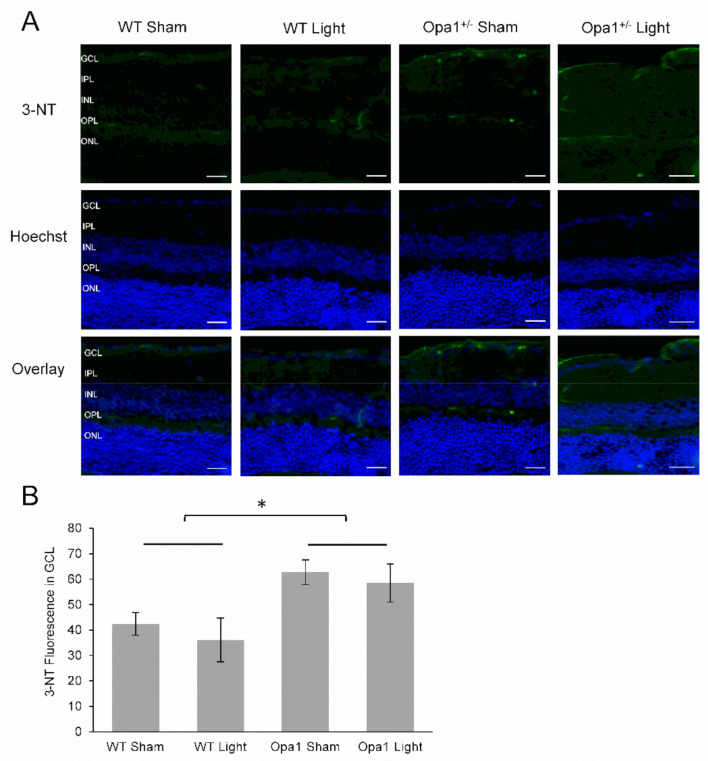
Aged Opa1^+/−^ mice had increased levels of 3-Nitrotyrosine in the GCL. (**A**) Representative images from 3-nitrotyrosine (3-NT) immunostaining. The top row shows 3-NT staining in green, the middle row is a nuclear stain, and the bottom row shows the overlay of the two stains. GCL: ganglion cell layer; IPL: inner plexiform layer; INL: inner nuclear layer; OPL: outer plexiform layer; ONL: outer nuclear layer. Scale bar represents 20 µm. (**B**) Quantification of fluorescence showed significantly increased levels of 3-NT in Opa1^+/−^ mice compared to WT controls. WT Sham (*n* = 3), WT Light (*n* = 4), Opa1^+/−^ Sham (*n* = 4), Opa1^+/−^ Light (*n* = 2). * *p* < 0.05, two-way ANOVA.

**Figure 5 ijms-22-08380-f005:**
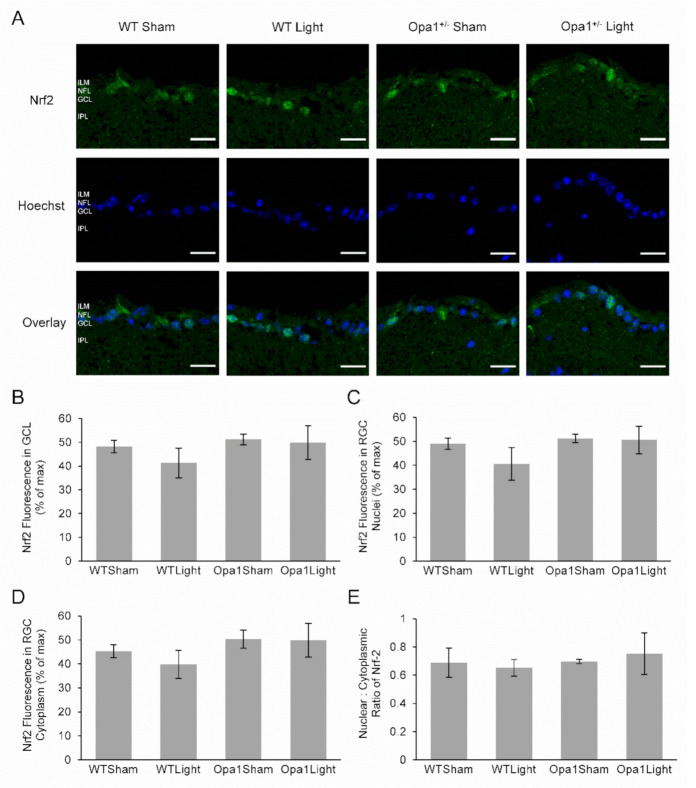
Immunohistochemical analysis of Nrf-2 expression. (**A**) Representative microscope images showing cross-sections of the GCL at 60× magnification. Nrf-2 staining is shown in the top row, while the middle row shows the nuclear stain, and the bottom row contains overlays of the two stains. ILM: inner limiting membrane; NFL: nerve fiber layer; GCL: ganglion cell layer; IPL: inner plexiform layer. Scale bar represents 20 µm. (**B**–**E**) The Nrf2 fluorescence was quantified over the whole of the GCL, as well as specifically within the nuclear and cytoplasmic compartments in order to analyze the nuclear: cytoplasmic ratio as a measurement of Nrf2 activation. WT Sham (*n* = 3), WT Light (*n* = 5), Opa1^+/−^ Sham (*n* = 5) and Opa1^+/−^ Light (*n* = 3).

**Figure 6 ijms-22-08380-f006:**
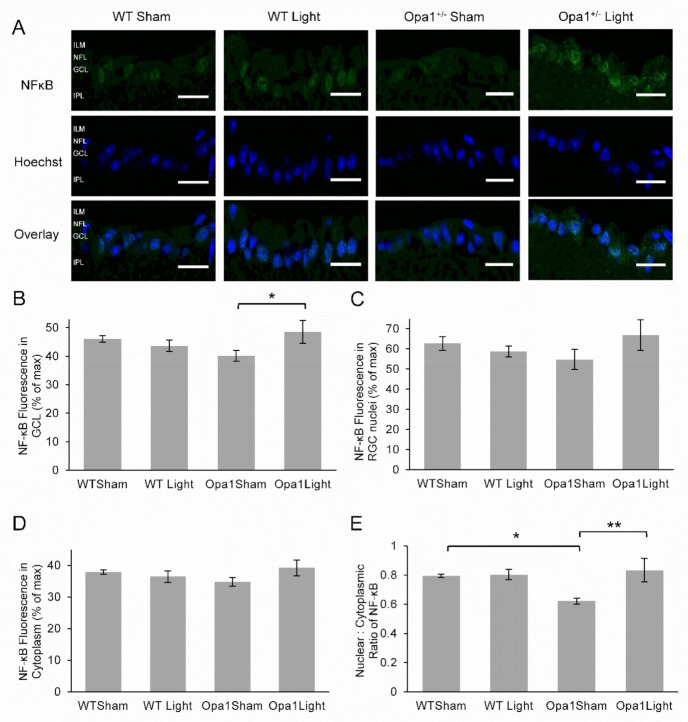
Immunohistochemical analysis of NF-κB expression in the GCL. (**A**) Representative images of the GCL taken at 60× magnification. The top row shows NF-κB staining in green, the middle rows depict a blue nuclear stain, and the bottom row displays the overlay. ILM: inner limiting membrane; NFL: nerve fiber layer; GCL: ganglion cell layer; IPL: inner plexiform layer. Scale bar represents 20 µm. (**B**) The mean fluorescence was quantified in the GLC and showed no significant differences. (**C**) NF-κB fluorescence was reduced in the GCL nuclei of light-treated WT mice compared to sham-treated WT controls. (**D**) Cytoplasmic NF-κB was reduced in the GCL of both sham-treated Opa1^+/−^ and light-treated WT mice compared to WT controls. (**E**) The nuclear: cytoplasmic ratio, representing NF-κB activation, was significantly decreased in the Opa1^+/−^ Sham group compared to WT controls. However, this decrease was not observed in mutant mice pre-treated with 670 nm light. WT Sham (*n* = 3), WT Light (*n* = 6), Opa1^+/−^ Sham (*n* = 4) and Opa1^+/−^ Light (*n* = 3). * *p* < 0.05, ** *p* < 0.01 two-way ANOVA.

**Figure 7 ijms-22-08380-f007:**
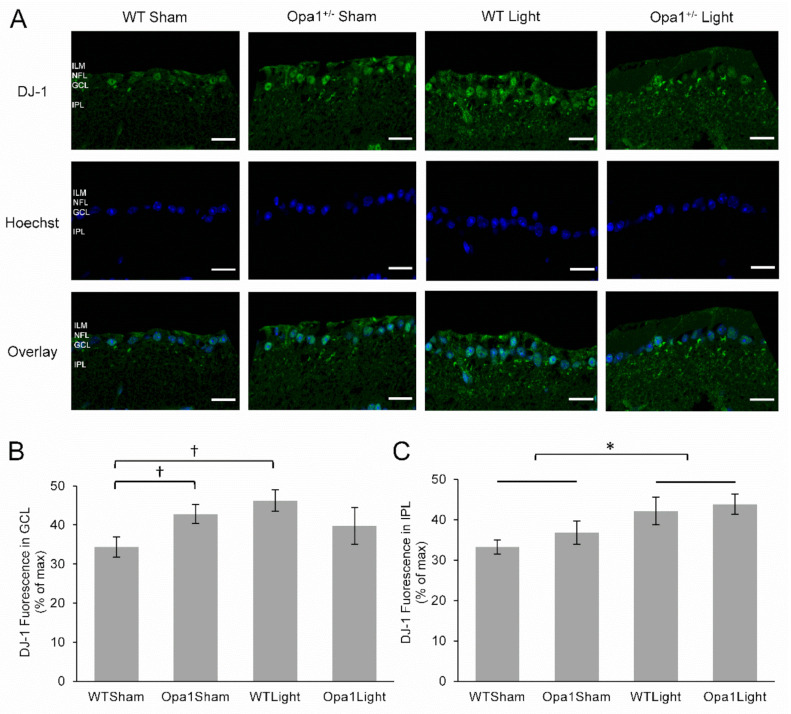
DJ-1 expression was upregulated in the IPL Following PBM: (**A**) Representative cross-sections of the GCL and IPL taken at 60× magnification. The top row shows DJ-1 staining in green, the middle row shows the nuclear stain, and the bottom row shows an overlay of the two stains. ILM: inner limiting membrane; NFL: nerve fiber layer; GCL: ganglion cell layer; IPL: inner plexiform layer. Scale bar represents 20 µm. (**B**) Quantification of DJ-1 fluorescence revealed upregulation in the GCL of retinas from sham-treated Opa1^+/−^ mice and light-treated WT mice when compared to the sham-treated WT group. (**C**) There was upregulation of DJ-1 in the IPL in response to 670 nm light treatment in both WT and Opa1^+/−^ mice compared to their sham-treated counterparts. WT Sham (*n* = 3), WT Light (*n* = 7), Opa1^+/−^ Sham (*n* = 5) and Opa1^+/−^ Light (*n* = 3). * *p* < 0.05 main effect of treatment, two-way ANOVA; † *p* < 0.05, Mann-Whitney U.

**Figure 8 ijms-22-08380-f008:**
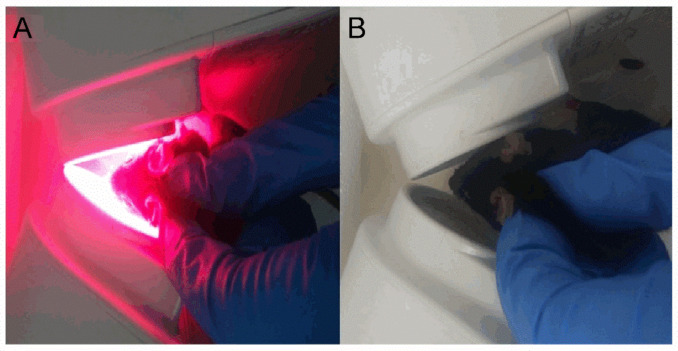
Irradiation procedure. (**A**) Light-treated mice were scruffed in front of two WARP 10 light devices for 88 s, positioned as shown, to ensure that an irradiance of approximately 50 mW/cm^2^ reached the cornea. (**B**) Sham-treated mice were scruffed in front of the light source for 88 s to ensure the level of stress was uniform throughout the experimental groups.

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
