# Peer review of "Red Light Irradiation In Vivo Upregulates DJ-1 in the Retinal Ganglion Cell Layer and Protects against Axotomy-Related Dendritic Pruning"

_ijms, 2021, doi:10.3390/ijms22168380_

Round 1
Reviewer 1 Report
Birne. et al. reported that red light irradiation in vivo can protect RGCs from axotomy-related dendritic pruning through DJ-1 upregulation. It is unclear the origin of this protection of RGCs from red light irradiation?
- IHC results focused on RGC layer, but they are not obvious in Figures 4, 5, 6, and 7. It would be nice if authors can provide immunoblotting (western blot) to show the changes in the DJ-1, NFkB, and Nrf2.
- Line 221-223, Figure 5, it is unclear the translocation of Nrf2 from the cytoplasm to the nucleus.
- Line 326-330, it is unclear how authors define “modest” generation of ROS in this experiment.
- Figure 8, the brightness of the irradiation is quite strong, are there any potential phototoxicity for mice and future human clinical trial?
Reviewer 2 Report
The manuscript of Beirne et al. "Red light irradiation in vivo upregulates DJ-1 in retinal ganglion cells and protects against axotomy-related dendritic prun ing" describes an interesting protection mechanism for RGC-death in OPA-1 mice.
Although the approach is very interesting and the evaluation is sound, the manuscript lacks further data to prove the hypothesis:
- The authors should test another wavelength besides red light to prove that it is red-light dependant and not just a reaction to intensifying light.
- Another quantitative method is needed to support the increased survival of RGCs e.g. RT-PCR, WB, but also MEAs would be OK.
- Same for the stress markers: a quantitative method is needed here: RT-PCR, ELISA, WB, or some of the many available assays.
- Next, activated downstrom factors of DJ-1 would support the hypothesis.
- Longer incubation times with red light and more treatment days should be tested. Maybe better effects can be achieved.
Minor comments:
Headlines in the results part should describe the results and not just name a marker.
Why did the authors choose 88 seconds for treatment? And why only for five days?
The authors should briefly describe their DiOlistic labelling in the method section, as this is the major point of their study.
The authors should not ignore the many publications on the OPA1 mouse from the group of Prof. Bernd Wissinger.
The groups on the x-axis should always be presented in the same order. Figure 7 is different from the rest.
Reviewer 3 Report
Manuscript Summary:
- The authors are investigating the in vivo pre-treatment with 670 nm light to assess the prevention of RGC degeneration over 16 hours following axotomy. They established that OPA1 resulted in increased nitrosative damage in the ganglion cell layer (GCL) alongside reduced NF-kB activation and upregulated DJ-1. They identify that PBM restored NF-κB 24 activation in Opa1+/- RGCs and enhanced DJ-1 expression in both genotypes after treatment, indicating a potential molecular mechanism within the retina to resist future oxidative insult. Even though, the manuscript addresses a relevant scientific question and pertinent studies, but major comments need to be addressed to enhance this paper.
Comments
- The authors should have more background on the connection of mitochondria with Photobiomodulation (PBM) and why that would impact it in terms of reactive oxygen species in the Introduction.
- No differences were seemingly observed between Figs. 2C and D. It seems like WT and OPA1 mutant mice had similar patterns of dendritic arborization. Also, in both “Number of intersections” graphs for Figures 2 and 3, the WT sham and light should be compared on the same graph and so should OPA1 sham and light. This would allow the reader to directly compare the effect of PBM on these mice with and without treatment.
- Dendritic arborization was quantified by three measures: the area under the Sholl curve (AUC), (Figure 2 E) the maximum peak of the Sholl curve (Figure 2 F), and the average total dendritic length (Figure 2 G). However, no significant differences were observed, an extended time course could have potentially increased degeneration.
- There was no significant amelioration of even the ON-center RGC dendropathy between the OPA1 sham and light in all different modes of evaluation including the AUC, Sholl analysis and dendritic length. Justification needs to be given on why the dendropathy could not be rescued by PBM compared to sham.
- The authors mention, ‘pre-treatment with 670 nm light significantly inhibited degeneration in the total population of WT RGCs, as the AUC, Sholl peak and dendritic length were all significantly greater in the light-treated cells compared to sham-treated after 16 hours in culture following axotomy”. And “while significant degeneration was observed over 16 hours following axotomy in ON-center RGCs from both WT and Opa1+/- sham-treated groups, it did not occur in RGCs that had been pre-treated with 670 nm light.” These statements were not depicted on the graphs and data presented. No significant rescue was observed with light compared to sham.
- For the 3-nitrotyrosine (3-NT) immunostaining, why were the WT and OPA1 mice not age matched? Further, Figure 4A does not show co-labeling with a RGC marker, so it is hard to discern if the positive staining is from RGCs or displaced amacrine cells in the GCL. Similarly for Figures 5A and 6A, RGCs need to be identified with Nrf-2 and NF-κB staining.
- The graphs for Figures 6B and C are depicting the same expression pattern of NF-κB. What is the difference between both the graphs? Basically, in terms of GCL, why have the authors made a distinction in staining of GCL and nucleus of RGCs? Is the staining in the GCL also depicting displaced amacrine cells? Also, there would be no need to present Figures 6C and D as Figure 6E is depicting the ratio differences and the importance of the light treatment in NF-κB activation in the OPA1 mutant mice. Same goes for Figures 5C and D.
- In Figure 7A, clearer pictures of DJ-1 need to be presented as there is a lot of background staining which is not a true depiction of DJ-1 expression as a transcription factor.
- The authors in discussion mentioned “Interestingly, over the course of 16 hours there was no significant progression of RGC dendropathy in Opa1+/- retinas.” So why did the authors not extend the time-course?
- The authors state the statement, “670 nm light pre-treatment over a period of 5 days neither reversed the loss of RGC dendritic complexity, nor reduced 3-NT levels in aged Opa1+/- retinas, despite its previous association with alleviation of nitrosative stress [52]”. However, this reference is for transected spinal cord slices undergoing secondary degeneration. Additionally, if this has already been documented, why are the authors reassessing the effects of PBM in short time-courses.
- The analyses were not performed masked, or the authors should detail the rigor within the experiments.
- In addition, within the methods section, the specific “n’s” of each experimental cohort have not been mentioned. Majorly some of the results are not statistically significant. Is there a possibility that this is due to the reason that a small number of n’s have been used for each experimental group with each group sometimes only being a n of 3 or 4?
Round 2
Reviewer 1 Report
I am not sure/convinced the nuclear translocation or location of the green fluorescence in Figure 5 and 6.
Author Response
Activation of Nrf2 and NFkappaB is associated with their translocation from the cytoplasm to the nucleus. Therefore we quantified the cytoplasmic and nuclear staining and their ratio as a measure of the activation of the respective transcription factors as has been published previously (Noursadeghi et al 2008). Figures 5 and 6 have the ganglion cell layer labelled, showing the location of the green fluorescence and blue fluorescence of the nuclear stain. The bottom panel shows the merge of the blue and green stains allowing visualisation of where the green fluorescence localises with the blue, giving an indication of the proportion of green fluorescence that is in the nuclear region compared to cytoplasm. Noursadeghi M, Tsang J, Haustein T, Miller RF, Chain BM, Katz DR. Quantitative imaging assay for NF-kappaB nuclear translocation in primary human macrophages. J Immunol Methods. 2008;329(1-2):194-200. doi:10.1016/j.jim.2007.10.015Reviewer 2 Report
The authors replied to all my questions raised. However, they did not perform any demanded experiments. Although I understand why the authors did not perform the experiments, they would have surely helped to increase the manuscript's impact.
With the current experiments, the impact of the manuscript is sadly rather limited.
Author Response
We thank the reviewer for their comments. We agree that additional experiments allowing to elucidate the role of DJ-1 in dendritic pruning would certainly increase the impact and we will pursue that investigation in our future research.
Reviewer 3 Report
The comments have been addressed by the authors.
Author Response
We thank the reviewer for their comments.